# Identification of a pyrophosphate-dependent kinase and its donor selectivity determinants

Ryuhei Nagata[1], Masahiro Fujihashi[1], Takaaki Sato[2], Haruyuki Atomi[2] & Kunio Miki[1]

Almost all kinases utilize ATP as their phosphate donor, while a few kinases utilize pyrophosphate (PPi) instead. PPi-dependent kinases are often homologous to their ATP-dependent counterparts, but determinants of their different donor specificities remain unclear. We identify a PPi-dependent member of the ribokinase family, which differs from known PPi-dependent kinases, and elucidate its PPi-binding mode based on the crystal structures. Structural comparison and sequence alignment reveal five important residues: three basic residues specifically recognizing PPi and two large hydrophobic residues occluding a part of the ATP-binding pocket. Two of the three basic residues adapt a conserved motif of the ribokinase family for the PPi binding. Using these five key residues as a signature pattern, we discover additional PPi-specific members of the ribokinase family, and thus conclude that these residues are the determinants of PPi-specific binding. Introduction of these residues may enable transformation of ATP-dependent ribokinase family members into PPi-dependent enzymes.

[1] Department of Chemistry, Graduate School of Science, Kyoto University, Sakyo-ku, Kyoto 606-8502, Japan. [2] Department of Synthetic Chemistry and Biological Chemistry, Graduate School of Engineering, Kyoto University, Katsura, Nishikyo-ku, Kyoto 615-8510, Japan. Correspondence and requests for materials should be addressed to M.F. (email: mfuji@kuchem.kyoto-u.ac.jp) or to K.M. (email: miki@kuchem.kyoto-u.ac.jp)

Kinases transfer a phosphate group from a phosphate donor to an acceptor. Various compounds are known as phosphate acceptors (e.g., proteins, lipids, and carbohydrates), while the donor for most kinases is ATP. A few kinases utilize atypical donors other than ATP. One utilizes ADP as the donor[1–4], another uses inorganic pyrophosphate (PPi)[5–7], and enzymes that utilize inorganic polyphosphate are also known[8,9]. The mechanism governing donor specificity and the evolutional trajectory of these unique enzymes have been widely discussed but are still unclear[1,5,6,8–12].

Only three types of enzyme have been identified as PPi-dependent kinases: PPi-dependent phosphofructokinase (PPi-PFK), PPi-dependent pyruvate phosphate dikinase (PPi-PPDK), and PPi-dependent acetate kinase (PPi-ACK). PPi-PFK catalyzes the phosphorylation of D-fructose 6-phosphate using PPi to produce D-fructose 1,6-bisphosphate, and also catalyzes the reverse reaction[11,13]. PPi-PPDK catalyzes the reversible conversion of phospho(enol)pyruvate, PPi, and AMP into pyruvate, orthophosphate (Pi), and ATP[13]. PPi-ACK, which has been found only in *Entamoeba histolytica*, catalyzes the reversible phosphorylation of acetate with PPi as the phosphate donor. However, PPi-ACK from *E. histolytica* is thought to primarily produce PPi and acetate from Pi and acetyl phosphate under physiological conditions because the $k_{cat}$ value of the PPi-producing reaction is 1000-fold higher than that of the PPi-consuming reaction[7]. The former two reactions are involved in glycolysis/gluconeogenesis[13], while PPi-ACK is presumed to provide PPi for PPi-PFK and PPi-PPDK during glycolysis in *E. histolytica*[7,14].

The reason why the PPi-dependent kinases prefer PPi to ATP as the phosphate donor remains unclear. The mechanisms of the donor specificity in PPi-PFK and PPi-ACK have been discussed without reference to their PPi-complex structures, but based only on structural comparisons with their ATP-dependent homologs. For example, the aspartate residue at the phosphate-donor-binding site in PPi-PFK prevents the ATP binding[6,15,16]. In PPi-ACK from *E. histolytica*, the five residues in the donor-binding site occlude the ATP-binding cleft[17,18]. Thus, only mechanisms for interfering with ATP binding have been suggested, while the residues that specifically recognize PPi remain unclear.

Here, we identify a PPi-dependent kinase belonging to the ribokinase family, which is distinct from the families of the previously reported PPi-dependent kinases. The crystal structure complexed with a PPi analog reveals the PPi-binding mode of this enzyme. Structural comparison and sequence alignment with ATP-dependent or ADP-dependent members of the ribokinase family reveal the importance of five residues: two large hydrophobic residues occluding a part of the ATP-binding pocket and three basic residues specifically involved in PPi recognition. The five residues are used collectively as a signature pattern and enable us to newly identify PPi-specific members of the ribokinase family.

## Results

**Identification of a PPi-dependent kinase.** A PPi-dependent kinase belonging to the ribokinase family was identified based on structural similarity to a *myo*-inositol 3-kinase from the hyperthermophilic archaeon *Thermococcus kodakarensis* (MI3K_TK), which is an ATP-dependent member of the ribokinase family[19,20]. A Dali search[21] with the substrate-complex structure of MI3K_TK (Protein Data Bank (PDB) ID 4XF7) showed that the structure is the most similar to the unliganded structure of TM0415 from the hyperthermophilic bacterium *Thermotoga maritima* (PDB ID 1VK4; root-mean-square (RMS) distance 2.2 Å for 254 Cα atoms out of 283 Cα atoms of TM0415; Fig. 1a). This enzyme has been annotated as a carbohydrate kinase belonging to the ribokinase family and is thought to be involved in *myo*-inositol metabolism because its gene is located in a *myo*-inositol catabolic gene cluster[22,23]. However, this enzyme exhibited no ATP-dependent kinase activity toward various carbohydrates, including *myo*-inositol[22,23]. Consistent with this result, a part of the potential ATP-binding cleft in TM0415 is occluded by three large residues (F221, R232, and M266; Fig. 1b). In contrast, comparison of the acceptor-binding site between TM0415 and MI3K_TK indicated that the five residues interacting with *myo*-inositol in MI3K_TK are conserved in TM0415 (Fig. 1c), raising the possibility that *myo*-inositol is the phosphate acceptor of TM0415. Accordingly, the phosphate donor specificity of TM0415 was investigated using *myo*-inositol as the acceptor. This analysis demonstrated that TM0415 utilizes PPi but neither ATP nor ADP to generate *myo*-inositol monophosphate (Fig. 2a). Although the ribokinase family includes various ATP-dependent or ADP-dependent kinases, including MI3K_TK, no member has been shown to be PPi-dependent until now.

Next, we performed kinetic analyses of TM0415 toward PPi and *myo*-inositol (Supplementary Fig. 1a). The initial velocity of the TM0415 reaction was almost constant (~17 μmol min$^{-1}$ mg$^{-1}$) at a PPi concentration ranging from 15 to 500 μM. At lower concentrations (<15 μM), the experimental signal to determine the initial velocity was lower than the detectable limit. From the

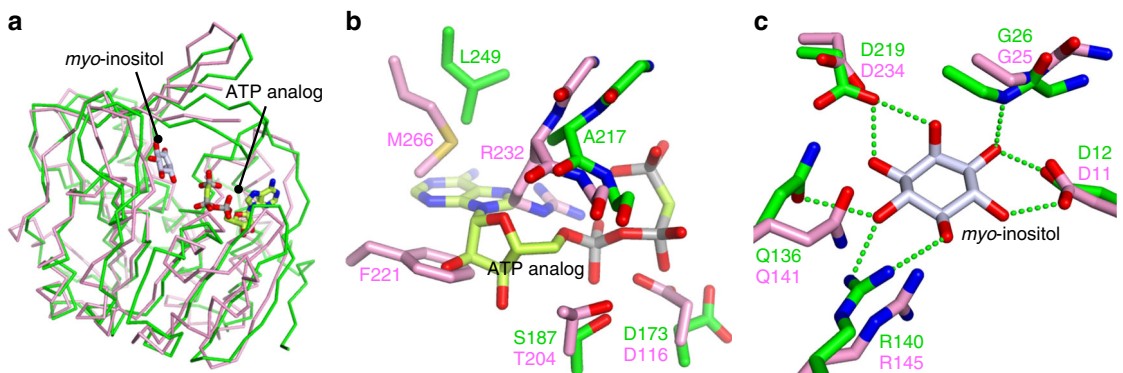

**Fig. 1** Structural comparison between TM0415 and MI3K_TK. The structures of the unliganded TM0415 (PDB ID 1VK4) and the substrate complex of MI3K_TK (4XF7) are shown in pink and green, respectively. **a** Superposition of the Cα traces of the overall structures. **b,c** Comparison of the phosphate-donor- or acceptor-binding site. Green dotted lines represent the interactions between *myo*-inositol and the residues in MI3K_TK. The ligands and residues in the ligand-binding site are shown as sticks

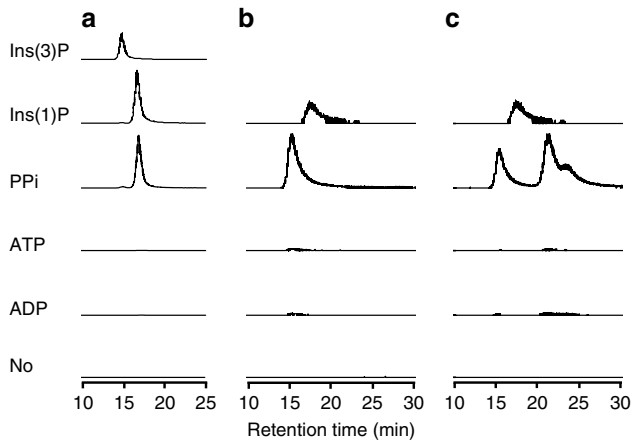

**Fig. 2** Analyses of the phosphate donor and the product. LC-MS analyses on the *myo*-inositol monophosphate produced from the reactions with TM0415 (**a**), homolog No. 7 (**b**), and homolog No. 49 (**c**). Ins(1)P and Ins(3)P represent the authentic compounds, and "No" means that the reaction was performed without phosphate donors. Eluted *myo*-inositol monophosphate was detected by MS with a mass range of *m/z* 259.0211–259.0237

results, the values of $K_m$ and $k_{cat}$ toward PPi were estimated to be <15 μM and ~9.5 s$^{-1}$, respectively. The estimated $K_m$ value is comparable to those of previously reported PPi-dependent kinases (1.2–200 μM)[6,7,12,24–30]. On the other hand, the values of $K_m$ and $k_{cat}$ toward *myo*-inositol are 12 ± 2 mM and 10.8 ± 0.3 s$^{-1}$, respectively. The $K_m$ value is 40 times higher than that of MI3K_TK (0.30 ± 0.03 mM)[19]. This raises the possibility that the genuine phosphate acceptor of TM0415 is a compound other than *myo*-inositol: for example, compounds related with the *myo*-inositol metabolic pathway that is composed of enzymes encoded by the TM0411–TM0416 operon in *T. maritima*, such as 2-keto-*myo*-inositol, diketo-inositol, 5-keto-L-gluconate, and D-tagaturonate (Supplementary Fig. 1b)[23]. These compounds have many hydroxyl and carbonyl groups, and thus possibly make hydrogen bonds with residues in the acceptor-binding site of TM0415. The $k_{cat}$ value toward *myo*-inositol is approximately the same as that toward PPi, verifying that the kinetic analyses of the two substrates were carried out accurately.

The effect of a potassium ion on the TM0415 activity was investigated because the activation of some ribokinase family enzymes by monovalent cations has been reported[31–33]. However, activation of the TM0415 reaction was not observed while adding KCl up to 100 mM (Supplementary Fig. 1a). The lack of activation of TM0415 by the monovalent cation is consistent with one of its structural features. That is, structural comparison between TM0415 and ribokinase from *Escherichia coli* (RK_EC, PDB ID 1GQT), which is activated by potassium or cesium ions[31], showed that the amino group of K265 in TM0415 occupies the position of the monovalent cation in RK_EC (Supplementary Fig. 2). MI3K_TK, which is not activated by the potassium ion, also possesses R252, which occupies the corresponding position[20]. The presence of these positive residues probably prevents the monovalent cations from binding and results in the insensitivity of TM0415 to the monovalent cations.

**Phosphate-donor recognition**. In order to elucidate the PPi-binding mode of TM0415, we determined its crystal structure complexed with a PPi analog (methylenediphosphonic acid: PCP), which has a carbon atom instead of an oxygen atom in the bridge position of PPi. In an asymmetric unit, two protein molecules were found: chains A and B (Supplementary Fig. 3a).

The dimer structure probably does not reflect the physiological form of the protein, because size exclusion chromatography showed that TM0415 is a monomeric protein in solution (Supplementary Fig. 3b). Chain A was complexed with PCP, a magnesium ion, and *myo*-inositol, while chain B possessed a sulfate ion, which is derived from a crystallization precipitant, instead of PCP without any magnesium ions (Supplementary Fig. 3a,c). The structures of chains A and B were very similar to each other (RMS distance 0.6 Å for 274 Cα atoms out of 275 chain B Cα atoms) and also similar to the unliganded structure (RMS distances 0.9 Å for 274 Cα atoms and 1.0 Å for 275 Cα atoms, respectively, out of 283 Cα atoms of the unliganded structure; Supplementary Fig. 3d). Slight differences were found only around the ligand-binding site.

In chain A, the phosphoryl group of PCP near *myo*-inositol (the proximal phosphoryl group) interacts with the main-chain nitrogen atoms of G231 and D234, while the other (the distal phosphoryl group) is surrounded by K171, T204, and R232 (Fig. 3a). The two phosphoryl groups and four water molecules octahedrally coordinate with the magnesium ion, which is essential for the kinase activity (Fig. 3b). R229 is also located near PCP, but its side chain was disordered in the structure (Fig. 3a). We thus also determined the SO$_4$$^{2-}$-complex structure, which possesses only the sulfate ions and *myo*-inositol in both molecules in an asymmetric unit. The sulfate ion is well superimposed on the distal phosphoryl group of PCP in a protein molecule that crystallographically corresponds to the PCP-bound chain described above (Fig. 3a). This suggests that its binding mode mimics the true binding mode of the distal phosphate group of PPi because the sulfate ion has no methylene group. This sulfate ion interacts with the side chain of R229 and the main-chain nitrogen atom of G233 in addition to K171, T204, and R232 (Fig. 3a). This strongly suggests that R229 and G233 recognize PPi in the reaction. The contribution of R229 to the reaction was confirmed by the fact that an R229A mutant exhibited only 7% of the level of the wild-type activity (Fig. 3b). The disorder of R229 in the PCP-complex structure is thought to result from the inability of the methylene group in PCP to form hydrogen bonds with the guanidine head of R229. The probable PPi-binding mode is depicted in Fig. 3c.

The conservation of the residues contributing to PPi recognition (Fig. 3c) was investigated in the ribokinase family enzymes. Sequence alignment based on structural superposition showed that K171, R229, and R232 of TM0415 are not conserved in the ATP-dependent and ADP-dependent members (Fig. 3d). We then thought that it might be possible to actually predict PPi-dependent kinases in the ribokinase family using these three characteristic residues as a part of a signature pattern. R232 is also one of the three large residues occluding the ATP-binding pocket, as described above (Fig. 1b). Mutation of each characteristic residue in TM0415 into alanine led to a drastic decrease in the specific activity (3–11% of the wild type; Fig. 3b), confirming that the three basic residues contribute to the reaction.

**Phosphate-acceptor binding**. The PCP-complex structure of TM0415 also revealed that its *myo*-inositol-binding mode is different from that of MI3K_TK, although the residues interacting with *myo*-inositol in MI3K_TK are conserved in TM0415 (Fig. 1c). D11, N78, Q141, and R145 in TM0415 interact with four of the six hydroxyl groups in *myo*-inositol (Fig. 4a), whereas D12, G26, Q136, R140, and D219 in MI3K_TK interact with all six hydroxyl groups (Fig. 4b)[19]. As shown in Fig. 4c, structural superposition of the two enzymes showed that the six-membered ring of *myo*-inositol tilts 30–40° and rotates ~120° from each other. One of the possible reasons for this shift is the difference of the hydrophobic residues in the *myo*-inositol-binding pocket:

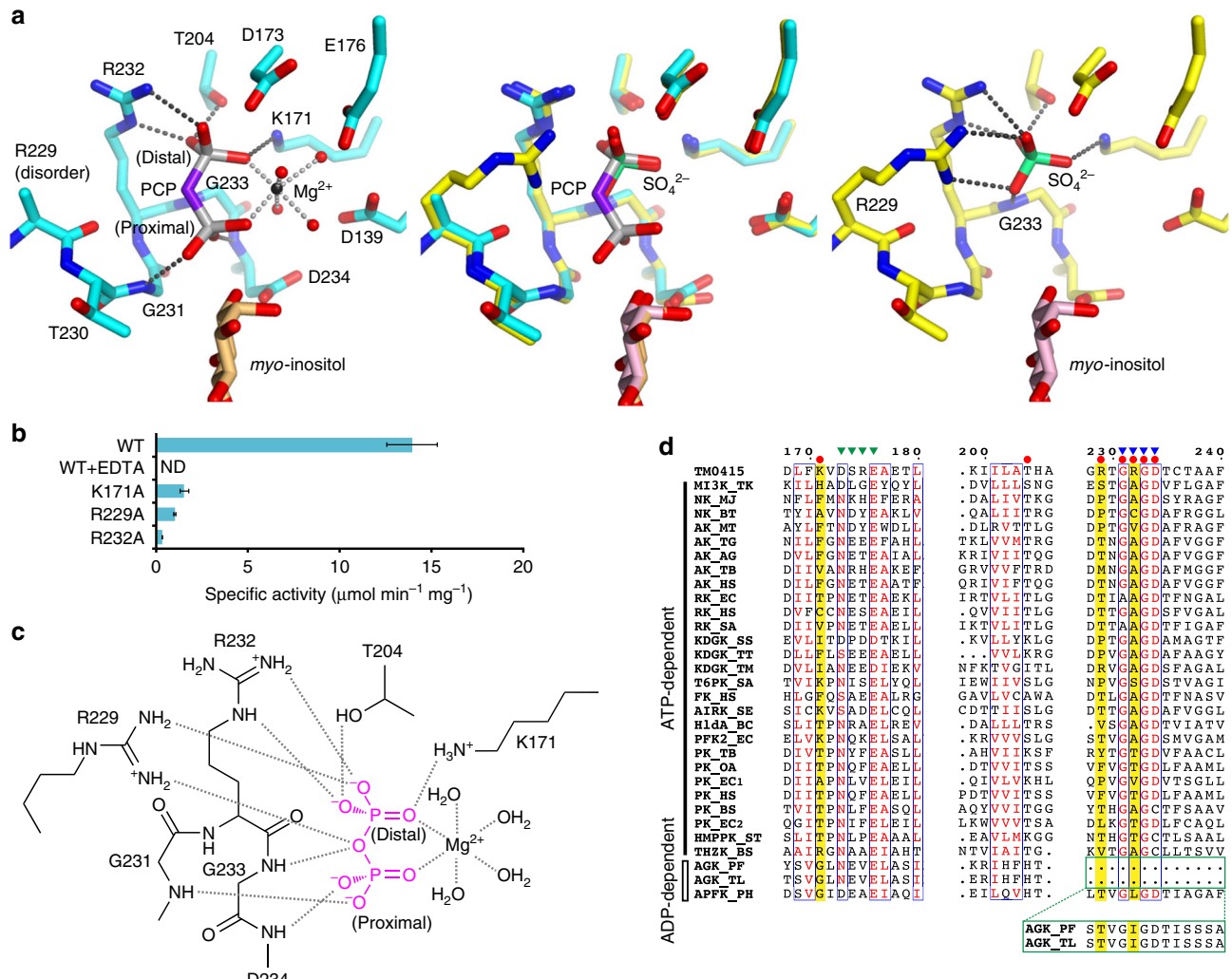

**Fig. 3** PPi recognition in TM0415. **a** The PPi-binding sites of the PCP complex (left, cyan) and the $SO_4^{2-}$ complex (right, yellow) of TM0415. Superposition of the two complexes is shown in a center panel. Dotted lines represent the interactions involving PCP, the sulfate ion, and the magnesium ion. The three panels are drawn from the same viewpoint. **b** Specific activities of TM0415 (wild-type and its mutants). WT, wild-type; WT + EDTA, wild-type in the presence of 1 mM EDTA. ND means no detectable activity. Activity measurements were performed in triplicate, and standard deviations are represented as error bars. **c** A schematic diagram depicting the probable PPi-recognition mode. Gray dotted lines show the interactions involving PPi (magenta) and the magnesium ion. **d** Sequence alignment of the ribokinase family enzymes based on 3D-structure superpositions. The sequences around the GXGD motif in AGK_PF and AGK_TL are displayed in a green box under the alignment, because the order of the secondary structures around the GXGD motif in the two AGKs is different from those in the other enzymes, although the regions around the motif are superimposable in 3D-structure. Details are explained in Supplementary Fig. 7. Red circles indicate the residues interacting with PPi in panel **c**. Yellow bars highlight the characteristic residues of TM0415 (K171, R229, and R232) and the corresponding residues. Green and blue triangles indicate the NXXE motif and GXGD motif, respectively. Abbreviations of the ribokinase family enzymes are shown in Supplementary Table 7

namely, L77, L87, and V112 in MI3K_TK are replaced with N78, S89, and L116 in TM0415, respectively (Fig. 4a,b).

The difference of the binding mode led to a difference of the phosphorylated position of the produced *myo*-inositol monophosphate. In the TM0415 structure, the probable catalytic residue D234 and the proximal phosphoryl group of PCP are nearest to the 1-hydroxyl group of *myo*-inositol (3.7 and 3.4 Å, respectively, Fig. 4a). In addition, HPLC analysis using a chiral column showed that the elution time of the TM0415 product coincided with that of 1D-*myo*-inositol 1-phosphate (Ins(1)P) but not that of 1D-*myo*-inositol 3-phosphate (Ins(3)P), which is the product of MI3K_TK[19] (Fig. 2a). The structural and chromatographic analyses led us to conclude that TM0415 phosphorylates the 1-hydroxyl group of *myo*-inositol and produces Ins(1)P.

**Seeking unidentified PPi-dependent kinases.** The three basic residues recognizing PPi (K171, R229, and R232; Fig. 3a,c) and the two large hydrophobic residues occluding the ATP-binding pocket (F221 and M266, which together with one of the three basic residues, R232, are the three large residues shown in Fig. 1b) in TM0415 are expected to be the key residues for discovering various unidentified PPi-dependent kinases from the ribokinase family. An initial BLAST search using the overall TM0415 sequence as the query found only 24 homologs possessing the five key residues. The hit numbers were increased to 52 homologs by submitting the PPi-binding domain (residues 169–286) of TM0415 as the query sequence. In general, the overall structure of the ribokinase family enzymes is divided into two domains: the phosphate-acceptor-binding domain (the N-terminal half) and the donor-binding domain (the C-terminal

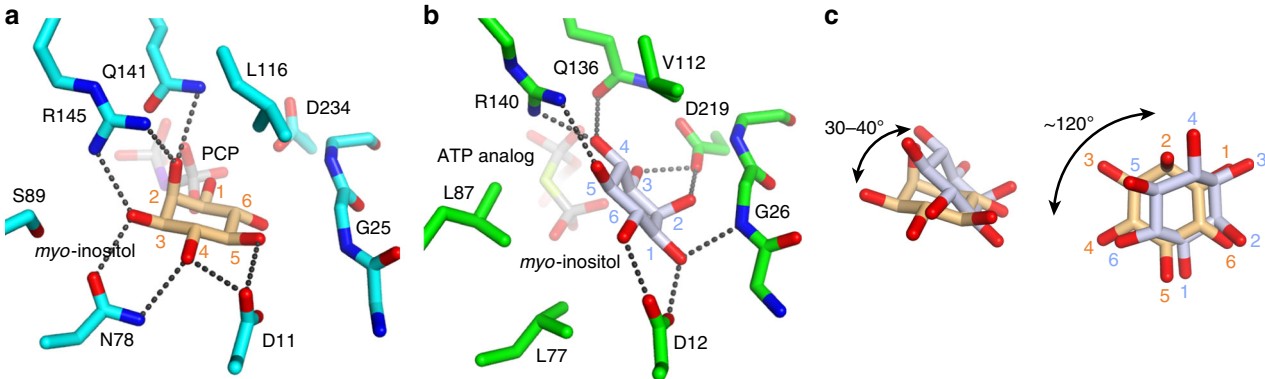

**Fig. 4** *myo*-inositol-binding mode. **a,b** The *myo*-inositol-binding sites in the PCP-complex of TM0415 (cyan) and the substrate complex of MI3K_TK (green, PDB ID 4XF7[19]). Dotted lines represent the interactions between *myo*-inositol and the residues. The two panels are drawn from the same viewpoint. **c** Structural superposition of the two *myo*-inositol molecules bound to TM0415 (light orange) and MI3K_TK (pale blue). Double-headed arrows indicate the rotation of *myo*-inositol. The two panels are drawn from two different viewpoints. The atom numbering of *myo*-inositol carbons is shown in the same color in panels **a** and **b**

half; Supplementary Fig. 3e). The increase in the hit number may result from elimination of the noise from the acceptor-binding domain in the BLAST search. Among the 52 homologs, two were TM0415 itself, and the other 50 homologs were presumed as candidate PPi-dependent kinases. Twenty of the candidate PPi-dependent kinases displayed no significant difference in the acceptor-binding site compared to TM0415 (Supplementary Fig. 4; Supplementary Table 1), suggesting that their phosphate acceptors are the same as that of TM0415. The other 32 homologs exhibited differences in the acceptor-binding site (Supplementary Table 2), raising the possibility that their acceptors are different from that of TM0415.

In order to examine whether the candidate PPi-dependent kinases exhibit PPi-specific kinase activity, recombinant proteins of five homologs (Nos. 7, 48, 49, 92, and 111) among the 32 homologs in Supplementary Table 2 were prepared by heterologous expression in *E. coli*. There were some differences in the sequences of the acceptor-binding sites among the five homologs. The expression of homolog No. 92 was poor (Supplementary Fig. 5a), and thus the activities of the other four homologs were investigated. Fourteen candidates for acceptors were tested: pentose (D-ribose and D-xylose), hexose (D-fructose and D-glucose), amino sugar (D-glucosamine), sugar alcohol (glycerol, *meso*-erythritol, and *myo*-inositol), disaccharide (sucrose and maltose), nucleoside (inosine, adenosine, and cytidine), and 2-keto-3-deoxygluconate. The activities were assayed using malachite green, which allows photometric determination of the Pi concentrations. The assay showed that homolog No. 49, in addition to TM0415, produced Pi upon incubation with PPi and *myo*-inositol (Supplementary Fig. 5b). In contrast, low levels of Pi production were observed with homolog No. 7 under the same conditions, and no activity was observed with the other two homologs, Nos. 48 and 111, at least with the phosphate acceptors examined here (Supplementary Fig. 5b). We further carried out incubation for longer periods of time with these three homologs, and an apparent production of Pi by homolog No. 7 was observed when *myo*-inositol was used as the acceptor (Supplementary Fig. 5c). No significant activity of the other two homologs was detected even with overnight (17 h) incubation (Supplementary Fig. 5c). LC-MS analysis confirmed that homologs No. 7 and 49 produce *myo*-inositol monophosphate, utilizing PPi but not ATP or ADP (Fig. 2b,c), strongly suggesting that these two homologs are PPi-dependent kinases. We presume that the absence of activities in homologs No. 48 and 111 may simply be due to the fact that the proteins mainly recognize

phosphate acceptors that differ from those applied in our experiments. The fact that two of the examined proteins actually exhibited PPi-dependent activity suggests that the five residues (K171, F221, R229, R232, and M266) in the donor-binding site can be used as signatures to predict and discover unidentified PPi-dependent kinases. Identification of the true substrates of homologs No. 48 and 111 and confirmation of PPi-dependent activity should greatly strengthen our proposal.

LC-MS analysis also provided insight into the phosphorylated position of the produced *myo*-inositol monophosphate by the homologs. The results clearly showed that the product of homolog No. 7 is not Ins(1)P (Fig. 2b). This change of product may arise from the replacement of I76 and S89 in TM0415 with cysteine and glutamine, respectively, in the acceptor-binding site (Supplementary Table 2). Homolog No. 49 seemed to produce three distinct *myo*-inositol monophosphates that exclude Ins(1)P (Fig. 2c). The product change may result from the replacement of N78, S89, and L116 in TM0415 with leucine, leucine, and serine, respectively, in the acceptor-binding site (Supplementary Table 2). The product diversity probably resulted from the ambiguous acceptor recognition. The non-specific recognition implies that the genuine acceptor of homolog No. 49 is a compound other than *myo*-inositol.

## Discussion

We discovered a PPi-dependent member of the ribokinase family, TM0415, and structurally elucidated the reason why the enzyme prefers PPi to ATP as the phosphate donor. The structural comparison between TM0415 and MI3K_TK showed that F221, R232, and M266 in TM0415 occupy a part of the ATP-binding cleft (Fig. 1b), suggesting that these three large residues prevent ATP from binding. The determined structures of TM0415 revealed the residues contributing to PPi binding (Fig. 3a,c). Three of them (K171, R229, and R232) are characteristic of TM0415 in the ribokinase family (Fig. 3d), and R232 is noteworthy for its contribution to both hindering ATP binding and interacting with PPi. The identification of characteristic residues of PPi recognition in PPi-dependent kinases has not been reported until now. In PPi-PFK and PPi-ACK, the PPi-recognition mode is unclear, and only residues obstructing ATP binding have been reported so far. In PPi-PFK, the conserved aspartate residue (e.g., D175 in the enzyme from *E. histolytica*) is suggested to prevent ATP binding, because its replacement with glycine, which is conserved in ATP-PFK, led to an 18-fold better

$K_m$ value toward ATP[6]. In PPi-ACK from *E. histolytica*, the five residues (T201, D322, Q323, M324, and E327) occlude the ATP-binding cleft of ATP-ACK[17,18]. These residues were introduced to ATP-dependent kinases in order to transform them into PPi-dependent ones, but these attempts were unsuccessful[6,17,18]. Our identification of the three basic residues of TM0415 may enable such transformation on ATP-dependent ribokinase family members, which phosphorylate various acceptors. This engineering will reduce costs for the production of a variety of phosphorylated compounds, because PPi is 1000-fold cheaper than ATP.

TM0415 and the 50-candidate PPi-dependent kinases may be members of a PPi-dependent group of the ribokinase family, which is the third subclass of this family in addition to those of the ATP-dependent and ADP-dependent enzymes. This is a unique example of a kinase family that contains ATP-, ADP-, and PPi-dependent enzymes. Thirty-two of the fifty enzymes exhibited some differences in the acceptor-binding site when compared to TM0415 (Supplementary Table 2). Two of them, homologs No. 7 and 49, phosphorylate *myo*-inositol utilizing PPi in the manner of TM0415, but the phosphorylated hydroxyl group is different from that of TM0415 (Fig. 2b,c). This change may result from the different binding orientation of the acceptor caused by substitutions in the acceptor-binding site. The other homologs in Supplementary Table 2 also display various substitutions in the acceptor-binding site, and thus the PPi-dependent kinases in the ribokinase family possibly include a variety of enzymes exhibiting different acceptor specificity.

Among the homologs examined in this study, homolog No. 49, which exhibited PPi-dependent kinase activity but no activity with ATP/ADP, is from *Levilinea saccharolytica*, a member of the phylum Chloroflexi in bacteria. Homolog No. 7, from *Acanthamoeba castellanii*, is a eukaryotic enzyme that also displayed activity with PPi. This suggests that the PPi-dependent ribokinase family members are not confined to the Thermotogae or bacteria. The source organisms of the 50-candidate PPi-dependent kinases are members of diverse bacterial phyla (e.g., Thermotogae, Proteobacteria, Spirochetes, and Chloroflexi) and three were from eukaryotic organisms, suggesting that the PPi-dependent ribokinase family enzymes may be widely distributed in nature.

R229 and R232 of the three basic residues in TM0415 are located around the GXGD motif. R229 is positioned two residues before the motif, and R232 corresponds to the second residue in the motif (Fig. 3d). This motif is conserved in the ribokinase family enzymes[33,34]. The aspartate residue is the catalytic residue, and the two glycine residues recognize the phosphate groups of the phosphate donor. The second residue is a small one (Ala, Cys, Val, Ser, or Thr) in ATP-dependent members to make room for the adenine and ribose groups of the nucleotide (Fig. 3d). In the ADP-dependent members, the residue is replaced with Ile or Leu to fill the small space created by the size difference between ADP and ATP. In the PPi-dependent kinase TM0415, the large residue R232 is situated in the corresponding position in order to recognize a smaller phosphate donor molecule. In addition, the two arginine residues are conserved in the 50-candidate PPi-dependent kinases (Supplementary Table 1, 2). Therefore, we propose an RXGRGD motif (the residues GRGD correspond to the GXGD motif) for the PPi-dependent kinases belonging to the ribokinase family.

The ribokinase family enzymes also possess an NXXE motif near the phosphate-donor-binding site, which is involved in magnesium binding[33,35]. In the PCP-complex structure, the magnesium-coordinated waters are surrounded by D139, D173, and E176 (Fig. 3a). The latter two residues correspond to the first and last residues of the NXXE motif. The asparagine residue in the motif is replaced with aspartic acid in TM0415. This substitution is also observed in some ATP-dependent enzymes in the

ribokinase family (Fig. 3d). Thus, the substitution of Asp for Asn in the NXXE motif may be unrelated to the specificity toward PPi.

As already described, three kinds of PPi-dependent kinases (PPi-PFK, PPi-PPDK, and PPi-ACK) have been identified thus far. We investigated whether the RXGRGD motif and/or the five key residues for the PPi-dependent members of the ribokinase family are found in the three kinds of enzymes. No signature patterns corresponding to the motif and the key residues were observed in PPi-PPDK and PPi-ACK. In the phosphate-donor-binding site of PPi-PFK, two basic residues were found (e.g., K148 and H384 in PPi-PFK from *Borrelia burgdorferi*), which are not conserved in ATP-PFK. However, their contributions to PPi binding remain to be elucidated. Further investigations including determination of the PPi-complex structure are necessary for identifying signature patterns for the three kinds of PPi-dependent kinases.

The evolutionary trajectory of the PPi-dependent kinases is a controversial topic. For example, three hypotheses of the evolutional relationship between PPi-PFK and ATP-PFK have been proposed: PPi-PFK evolved into or from ATP-PFK[5,6,11] or emerged from a common ancestor independently of ATP-PFK[36]. In the ribokinase family, the evolutional trace has been discussed based on the size of the lid domain[37], which covers the ligand-binding site. A kinase without a lid domain is thought to be the ancient type of kinase. All kinases without the lid domain are ATP-dependent, suggesting that this ribokinase family originated from an ATP-dependent enzyme without a lid. As the enzyme evolved, the lid domain occurred and became larger to protect its substrates from the solvent (Supplementary Fig. 6a). The size of the lid varies widely, ranging from those consisting of only loops (smallest) to those with five strands and four helices (largest). All reported ADP-dependent ribokinases harbor the largest lid domain[1], and are thus considered to have evolved from ATP-dependent enzymes with large, but slightly smaller lids with five strands and two helices (Supplementary Fig. 6a). TM0415 possesses a medium-sized lid domain with four β strands (Supplementary Fig. 6a). According to the hypothesis stated above, TM0415 is not at the root of the evolutionary tree of the ribokinase family enzymes. Primary sequences suggest that lid domains of similar size are found in most of the 50-candidate PPi-dependent kinases (Supplementary Fig. 6b). It should be noted that the relationship between PPi dependency and lid size is free from any query sequence bias. The query was composed of only the phosphate-donor-binding domain (the C-terminal half), whereas the lid domain is positioned in the N-terminal half. The conserved medium-sized lid domain implies that the PPi-dependent members of the ribokinase family emerged from the ancient ATP-dependent enzymes during the evolutional process.

## Methods

**Plasmid preparation.** Enzymes selected for activity measurements were prepared as N-terminal His-tag fusion proteins, while those for crystallization were produced without a His-tag. The *TM0415* gene was synthesized to produce a His-tag fusion protein (Supplementary Table 3) and inserted into the NcoI site of pET-15b by GenScript. The plasmids for expressing its mutants (K171A, R229A, and R232A) were prepared by inverse PCR using the TM0415 plasmid and the oligonucleotides K171A-F4, K171A-R4, R229A-F, R229A-R, R232A-F, and R232A-R (Supplementary Table 4) as a template and primers, respectively. In preparation of the enzymes for crystallization, the oligonucleotides (tag-rm-F and tag-rm-R; Supplementary Table 4) were used as primers for inverse PCR in order to remove the His-tag. The genes of the TM0415 homologs (No. 7, 48, 49, 92, and 111) were synthesized by GENEWIZ (Supplementary Table 5) and inserted into the NdeI and BamHI sites of the pCold II vector. The sequences of the resultant plasmids were confirmed by DNA sequencing (Hokkaido System Science or Macrogen Japan).

**Protein expression and purification.** *E. coli* strain BL21(DE3)pLysS (Novagen) cells were transformed with the plasmids described above. The transformants were cultured at 37 °C in lysogeny broth medium containing 100 μg/mL ampicillin and 100 μg/mL chloramphenicol. For gene expression using the pET-15b vector, iso-propyl-β-D-1-thiogalactopyranoside (IPTG) was added (final concentration 0.2

mM) at a cell density of 0.4 (optical density at 600 nm) to induce gene expression. After a further culture for 4 h, the cells were harvested by centrifugation (5,000×g for 15 min at 4 °C). In the expression with pCold II vector, the medium was cooled on ice for 30 min before the addition of IPTG. After the addition of IPTG and a further culture at 15 °C for 24 h, the cells were harvested by centrifugation.

The His-tagged TM0415 enzymes for the analysis of phosphate donor specificity were purified by Ni affinity chromatography. Cells were resuspended in buffer A (20 mM Tris-HCl (pH 7.4), 150 mM NaCl, 0.25 mM Tris(2-carboxyethyl) phosphine hydrochloride (TCEP-HCl)) containing an EDTA-free protease inhibitor cocktail (Nacalai Tesque) and disrupted by sonication. The sonicate was centrifuged (20,000×g for 30 min at 4 °C), and imidazole-HCl (pH 7.4) was added (final concentration 10 mM) into the supernatant. The supernatant was then mixed with Ni-NTA Superflow resin (QIAGEN) for 30–45 min at room temperature (RT). This mixture was loaded onto the column, and the flow-through fraction was collected. The resin was washed with buffer A supplemented with 50 mM imidazole-HCl (pH 7.4) for three column volumes (CVs), and the sample was eluted by buffer A supplemented with 300 mM imidazole-HCl (pH 7.4) for three CVs. The buffer of the eluate was exchanged with buffer A by ultrafiltration with an Amicon Ultra centrifugal filter unit (molecular weight cut off 10,000; Millipore).

The His-tagged TM0415 enzymes for kinetic analysis were purified by Ni affinity, anion exchange, and size exclusion chromatography. Cells were resuspended in buffer B (50 mM Tris-HCl (pH 7.9), 50 mM NaCl, 1 mM MgCl₂, 0.25 mM TCEP-HCl) and disrupted by sonication. The sonicate was centrifuged, and imidazole-HCl (pH 7.9) was added (final concentration 10 mM) into the supernatant. Affinity chromatography was performed as described above using the following buffers: wash buffer (50 mM potassium phosphate buffer (pH 7.8), 40 mM imidazole-HCl (pH 7.9), 300 mM NaCl, 10% (v/v) glycerol, 0.25 mM TCEP-HCl) and elution buffer (20 mM Tris-HCl (pH 7.9), 300 mM imidazole-HCl (pH 7.9), 10% (v/v) glycerol, 0.25 mM TCEP). The buffer of the eluted fractions was exchanged with buffer C (20 mM Tris-HCl (pH 8.1), 0.25 mM TCEP-HCl) by ultrafiltration. The sample was applied to a 1 mL MonoQ anion exchange column (GE Healthcare) equilibrated with buffer C and eluted with a linear gradient of 0–250 mM NaCl within 10 CVs. The eluted fractions at NaCl concentrations of 30–80 mM were concentrated by ultrafiltration. The sample was applied to a Superdex 200 Increase size exclusion column (GE Healthcare) equilibrated with buffer C supplemented with 150 mM NaCl and separated with the same buffer. The buffer of the relevant fractions was exchanged with 50 mM MES-NaOH (pH 6.1) for kinetic analysis.

Purification of the non-tagged TM0415 for crystallization was performed by heat treatment, anion exchange, and size exclusion chromatography. Cells were resuspended in buffer B supplemented with 10 mM MgCl₂ and disrupted by sonication. The sonicate was heat-treated at 80 °C for 15 min and centrifuged. In order to remove nucleic acid, the supernatant was treated with ~130 μg/mL deoxyribonuclease I from bovine pancreas (Sigma-Aldrich) and ~13 μg/mL ribonuclease A from bovine pancreas (Nacalai Tesque) for 90 min at RT. The buffer of this sample was exchanged with buffer C. The sample was further purified by anion exchange and size exclusion chromatography as described for the kinetic analysis, except that the eluted sample of the size exclusion column was just concentrated for crystallization.

TM0415 for acceptor screening was purified by Ni affinity chromatography alone. Cells were resuspended in buffer B and disrupted by sonication. The suspension was centrifuged, and the supernatant was purified by affinity chromatography. The chromatography was performed using the procedures applied for the purification of enzymes for kinetic analysis as described above. The buffer of the eluted samples of the affinity column was exchanged with buffer D (50 mM Tris-HCl (pH 7.9), 100 mM NaCl, 0.25 mM TCEP-HCl).

The homolog proteins were purified in the same way as TM0415 used for the acceptor screening, but disruption was performed by BugBuster Protein Extract Reagent (Novagen). After purification with the Ni-NTA column, the buffer of the sample was exchanged with buffer D or 100 mM ammonium acetate buffer (pH 6.6) by ultrafiltration for the acceptor screening or LC-MS analysis of the donor specificity, respectively.

**Acceptor screening of the TM0415 homologs**. Acceptor screening of homologs No. 7, 48, 49, and 111 was performed by the malachite green assay. The enzymatic reaction mixture (100 μL) was composed of 0.2 μg enzyme (TM0415 or its homologs), 350 or 7 mM phosphate acceptor, 500 μM PPi, 500 μM MgCl₂, 100 mM KCl, 100 mM NaCl, 0.25 mM TCEP-HCl, and 50 mM Tris-HCl (pH 7.9). The following phosphate acceptors were tested: D-ribose, D-xylose, D-fructose, D-glucose, D-glucosamine, glycerol, meso-erythritol, myo-inositol, sucrose, maltose, inosine, adenosine, cytidine, and 2-keto-3-deoxygluconate (the former 10 compounds were used at 350 mM and the others at 7 mM). These compounds were the representative acceptors of the ribokinase family enzymes except for the compounds affecting the malachite green coloring. After preincubation at 37 °C (or 70 °C for TM0415) for 3 min, the enzymatic reaction was initiated by adding PPi. The reaction was carried out for 10 min, 8 h, or 17 h and was terminated by cooling on ice for 5 min. An aliquot (50 μL) of the reaction mixture was blended with 100 μL of the BIOMOL GREEN reagent (Enzo Life Science) and incubated at RT for 20 min. The coloration was checked visually.

**LC-MS analysis on donor specificity and products**. Confirmation of the activities and analyses of the phosphate-donor specificities were performed by LC-MS. The reaction mixture with TM0415 (100 μL) was composed of 5 μg TM0415, 6 mM myo-inositol, 5 mM phosphate donor (ATP, ADP, or PPi), 10 mM MgCl₂, and 100 mM ammonium acetate buffer (pH 6.65). The reaction was carried out for 5 min at 85 °C. For the homologs, the reaction mixture (100 μL) was composed of 3 μg enzyme (homolog Nos. 7 and 49), 50 mM myo-inositol, 500 μM PPi, 500 μM MgCl₂, and 100 mM ammonium acetate buffer (pH 6.6). The reaction was performed for 17 h for 37 °C. The enzymes in the mixtures were removed by ultrafiltration, and the filtrate was analyzed by LC-MS. The methods used for the LC-MS analysis were reported previously[19]. Authentic Ins(1)P and Ins(3)P were purchased from Cayman Chemical Company.

**Kinetic analysis of TM0415**. Kinetic analysis of TM0415 was performed by detecting the produced Pi using a malachite green assay. The enzymatic reaction mixture (100 μL) was composed of 27 ng His-tagged TM0415, 15–500 μM PPi, 2–200 mM myo-inositol, 500 μM MgCl₂, and 50 mM MES-NaOH (pH 6.1). After preincubation at 70 °C for 3 min, the kinase reaction was initiated by adding PPi. The reaction was carried out at 70 °C for 1, 2, or 3 min and terminated by cooling on ice for 5 min. An aliquot (50 μL) of the reaction mixture was blended with 100 μL of the BIOMOL GREEN reagent and incubated at RT for 20 min. The absorbance of the mixture at 620 nm was measured using a V-630 spectrophotometer (JASCO Corporation). The amount of Pi produced in the kinase reaction was calculated from the absorbance based on the calibration curve. The kinetic parameters were determined using the program SigmaPlot (HULINKS Inc.).

Specific activities of the TM0415 mutants (K171A, R229A, and R232A) and wild type were also measured with the malachite green assay. The enzymatic reaction mixture (100 μL) was composed of 0.02–1 μg His-tagged enzymes, 500 μM PPi, 200 mM myo-inositol, 500 μM MgCl₂, and 50 mM MES-NaOH (pH 6.1). In these conditions, the enzymes seemed to exhibit $V_{max}$ from preliminary analysis. In analysis of the magnesium dependence, the mixture contained 1 mM EDTA instead of MgCl₂. The further procedure is the same as that of kinetic analysis described above.

**Crystallization and structure determination**. Crystallization was performed with the sitting-drop vapor diffusion method. The protein solution was composed of 10 mg/mL TM0415 (no His-tag, purified for crystallization), 500 mM myo-inositol, 10 mM PCP, 10 mM MgCl₂, 150 mM NaCl, 0.25 mM TCEP-HCl, and 20 mM Tris-HCl (pH 8.1). The solution was incubated at RT for 1 h and centrifuged (15,400×g for 5 min at RT) in order to remove the salt precipitate of PCP and magnesium. The supernatant was blended with an equal amount of the precipitant solution composed of 21–29% (w/v) poly(ethylene glycol) 4000, 2 or 20 mM ammonium sulfate (A/S), and 100 mM sodium acetate buffer (pH 4.6), and then equilibrated at 20 °C. The precipitant including 2 or 20 mM A/S was used for the co-crystallization with PCP or SO₄²⁻, respectively. The crystals were obtained within 1 month.

The crystals were soaked in cryo-protectant solution composed of 35% (w/v) poly(ethylene glycol) 4000, 2 or 20 mM A/S, 50 mM myo-inositol, and 100 mM sodium acetate buffer (pH 4.6) and flash-frozen in a nitrogen stream at 100 K. The concentration of A/S was the same as that of the precipitant solution of the crystals. Diffraction data sets were collected at the beamline BL41XU of SPring-8 at a wavelength of 1.000 Å. The data sets were integrated and scaled with the program HKL2000[38]. The phases were determined by the molecular replacement method with the atomic coordinates of the unliganded TM0415 (PDB ID 1VK4) using the program Molrep[39]. The structures were constructed using the program COOT[40] and refined using the program REFMAC5[41,42] with the Translation Libration Screw refinement technique. The statistics for data collections and refinements are summarized in Supplementary Table 6.

**Data availability**. The structural coordinates and structure factors have been deposited in the Protein Data Bank under accession codes 5YSP (the PCP-complex) and 5YSQ (the SO₄²⁻-complex). Other data are available from the corresponding authors upon reasonable request.

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

## Acknowledgements

We are grateful to the staff at the beamlines of the Photon Factory and SPring-8 for their help with the data collection and to Ms. Karin Nishimura for liquid chromatography mass spectrometry analysis. This work was supported by JSPS KAKENHI Grant Number 16J08482 (to R.N.) and 17H05439 (to M.F.). The use of beamlines at the Photon Factory and SPring-8 was approved by the Photon Factory Advisory Committee (2015G645 and 2017G696) and by the Japan Synchrotron Radiation Research Institute (JASRI; 2016A2743 and 2016B2723).

## Author contributions

R.N. designed the project, performed the experiments, and analyzed the results. M.F. designed the project and analyzed the results. T.S. performed biochemical assays and analyzed the results. H.A. and K.M. administered and supervised the project. All authors participated in writing of the paper.

## Additional information

**Competing interests:** The authors declare no competing interests.

