## [Peer Review File · Nature Communications]

Reviewers' comments:

Reviewer #1 (Remarks to the Author):

Nagata et al. report the identification of the first PPI-dependent member of the ribokinase family, the myo-inositol 3-kinase (TM0415) from the hyperthermophilic bacterium *Thermotoga maritima*. Kinetic analysis demonstrated that TM0415 utilized PPI as a phosphoryl group donor. Comparing the structures of the ATP-dependent myo-inositol 3-kinase from the hyperthermophilic archaeon *Thermococcus kodkarensis* with the unliganded structure of TM0415, the authors identified five residues that could be used to define PPI-dependent ribokinases. F221 and M266 are two large hydrophobic residues that occlude at least part of the ATP-binding pocket. K171, R229, and R232 are three basic residues that are involved in PPI recognition. Alteration of K171, R229, and R232 significantly reduced specific activity. Using these residues, the authors identified 24 possible PPI-dependent kinase homologs. 52 homologs were identified when using the PPI binding domain as the query. Of these homologs, recombinant proteins of five were produced and purified from *E. coli*. Using a variety of phosphoryl acceptors, only one of the homologs displayed activity with PPI, but not ADP or ATP, as phosphoryl donor.

Specific Comments/Questions:

(1) Page 2: In the last sentence of the second paragraph of the Introduction, the authors can add "...in *Entamoeba* sp.". The *Entamoeba* acetate kinase (Eh-Ack) is the only proven PPI-dependent Ack.

(2) Figure 3b: Did the authors kinetically analyze the K171A, R229A, and R232A variants? If so, what were the effects?

(3) Page 8: In the sentence ".....the five residues (T201, D322, G323, M324, and E327) occlude the ATP-binding cleft..." towards the end of the first paragraph of the Discussion, residue 323 in Eh-Ack is Gln (Q) not Gly (G).

(4) Page 8: Dang and Ingram-Smith altered Q323 and M324, two of the residues described in this paragraph. Their work (Dang, T. and C. Ingram-Smith (2017). "Investigation of pyrophosphate versus ATP substrate selection in the *Entamoeba histolytica* acetate kinase." *Sci Rep* 7(1): 5912.) should be noted and cited in this section. There are a few other places (e.g., sentences comparing ATP and PPI pockets) in the manuscript in which this work could be cited.

Reviewer #2 (Remarks to the Author):

PPI-dependent kinases, although rare, have been known for quite some time. Thus far, there has not been a structure of PPI-dependent kinase in complex with its cofactor PPI (or its non-functional analog). As a result, much of what we know about PPI-dependent kinases is based on indirect evidence. In this work, the authors identified a PPI-dependent kinase in the ribokinase family. They determined the PPI-dependent kinase ternary complex structure including the enzyme, co-factor (analog) and (potential) substrate. This is the first time such a complex structure is determined and, as such, provides first glimpse of cofactor binding, represents novel and original finding. Enzymatic analysis provided strong indication that TM0415, which is a ribokinase, is indeed PPI dependent. TM0415 is therefore a new and unconventional member of the ribokinase family. Based on the structure analysis, the authors further identified 5 key residues which they considered to be the signature motif of PPI kinase. Using the signature motif candidate PPI kinases in the database were identified and experimental results showed at least two of the potential hits have PPI kinase activity.

The manuscript is very nicely written - smooth with a nice logic flow, easy to follow and very informative. Although the amount of data is not very large, the novelty of this work is exemplified

by the first ternary complex structure and the resulting signature motif which is useful for identifying other PPI kinases.

I have a few comments:

1. The structure shows that residues F221, R232 and M266 prevent entrance of ATP and other structural features defining PPI-dependent kinases. The authors also suggest a strategy to convert PPI-dependent kinase to ATP-dependent kinase (and vice versa) by site directed mutagenesis. It would go a long way if an example of inter-conversion could be demonstrated through mutagenesis and activity assay.
2. The authors found that the K_m value of substrate myo-inositol is very large ($\sim 12 \pm 2$ mM). The possible reason may be a derivative(s) of myo-inositol which is the real substrate. It would be nice to at least suggest a few possibilities, although experimental validation may be beyond the scope of this work.
3. The 5-residue "signature" motif is an important result. Have the authors considered to see if it also works on all known PPI kinases beyond ribokinase family?
4. Size exclusion result of TM0415 showing it is a monomer in solution is not shown. It should be included in Supplementary Information.
5. It is not clear to this reviewer why there is a separate box in Fig. 3D corresponding to AGK_PF and AGK_TL (by the way, "_" is missing).

Point-by-point reply (manuscript NCOMMS-17-32278A)

We are grateful for the valuable comments and suggestions from the reviewers. We revised our manuscript (NCOMMS-17-32278A) based on the reviewers' comments, and our answers to their comments are shown below in blue.

Reviewers' comments:

Reviewer #1 (Remarks to the Author):

Nagata et al. report the identification of the first PPI-dependent member of the ribokinase family, the myo-inositol 3-kinase (TM0415) from the hyperthermophilic bacterium *Thermotoga maritima*. Kinetic analysis demonstrated that TM0415 utilized PPI as a phosphoryl group donor. Comparing the structures of the ATP-dependent myo-inositol 3-kinase from the hyperthermophilic archaeon *Thermococcus kodkarensis* with the unliganded structure of TM0415, the authors identified five residues that could be used to define PPI-dependent ribokinases. F221 and M266 are two large hydrophobic residues that occlude at least part of the ATP-binding pocket. K171, R229, and R232 are three basic residues that are involved in PPI recognition. Alteration of K171, R229, and R232 significantly reduced specific activity. Using these residues, the authors identified 24 possible PPI-dependent kinase homologs. 52 homologs were identified when using the PPI binding domain as the query. Of these homologs, recombinant proteins of five were produced and purified from *E. coli*. Using a variety of phosphoryl acceptors, only one of the homologs displayed activity with PPI, but not ADP or ATP, as phosphoryl donor.

Specific Comments/Questions:

(1) Page 2: In the last sentence of the second paragraph of the Introduction, the authors can add "...in *Entamoeba* sp.". The *Entamoeba* acetate kinase (Eh-Ack) is the only proven PPI-dependent Ack.

We clearly described that PPI-ACK is from *Entamoeba histolytica* in the second paragraph of Introduction on page 2.

(2) Figure 3b: Did the authors kinetically analyze the K171A, R229A, and R232A variants? If so, what were the effects?

We measured specific activity but did not perform intensive kinetic analysis of the TM0415 variants (K171A, R229A, and R232A), because we concentrated to judge whether these three residues are dominant for the PPI-dependent reaction or not.

Regarding to this comment, we added a method of the specific activity measurement on

page 14. As written in the method, the specific activity was measured at a PPi concentration of 500 μ M. The activity seemed to be close to V_{\max} based on a preliminary analysis: that is, initial velocities of all variants at the PPi concentration of 500 μ M are close to the highest value among those at 50, 500 and 1000 μ M.

(3) Page 8: In the sentence "...the five residues (T201, D322, G323, M324, and E327) occlude the ATP-binding cleft..." towards the end of the first paragraph of the Discussion, residue 323 in Eh-Ack is Gln (Q) not Gly (G).

Thank you for pointing this out. We changed G323 into Q323 at the 12th line from the bottom on page 8.

(4) Page 8: Dang and Ingram-Smith altered Q323 and M324, two of the the residues described in this paragraph. Their work (Dang, T. and C. Ingram-Smith (2017). "Investigation of pyrophosphate versus ATP substrate selection in the *Entamoeba histolytica* acetate kinase." *Sci Rep* 7(1): 5912.) should be noted and cited in this section. There a few other places (e.g., sentences comparing ATP and PPi pockets) in the manuscript in which this work could be cited.

As suggested by this reviewer, we added a new citation at the 5th line on page 3 and the 9th and 11th lines from the bottom on page 8.

Reviewer #2 (Remarks to the Author):

PPi-dependent kinases, although rare, have been known for quite some time. Thus far, there has not been a structure of PPi-dependent kinase in complex with its cofactor PPi (or its non-functional analog). As a result, much of what we know about PPi-dependent kinases is based on the indirect evidence. In this work, the authors identified a PPi-dependent kinase in the ribokinase family. They determined the PPi-dependent kinase ternary complex structure including the enzyme, co-factor (analog) and (potential) substrate. This is the first time such a complex structure is determined and, as such, provides first glimpse of cofactor binding, represents novel and original finding. Enzymatic analysis provided strong indication that TM0415, which is a ribokinase, is indeed PPi dependent. TM4014 is therefore a new and unconventional member of the ribokinase family. Based on the structure analysis, the authors further identified 5 key residues which they considered to be the signature motif of PPi kinase. Using the signature motif candidate PPi kinases in the database were identified and experimental results showed at least two of the potential hits have PPi kinase activity.

The manuscript is very nicely written - smooth with a nice logic flow, easy to follow and very

informative. Although the amount of data is not very large, the novelty of this work is exemplified by the first ternary complex structure and the resulting signature motif which is useful for identifying other PPi kinases.

I have a few comments:

1. The structure shows that residues F221, R232 and M266 prevent entrance of ATP and other structural features defining PPi-dependent kinases. The authors also suggest a strategy to convert PPi-dependent kinase to ATP-dependent kinase (and vice versa) by site directed mutagenesis. It would go a long way if an example of inter-conversion could be demonstrated through mutagenesis and activity assay.

As this reviewer commented, an example of the inter-conversion goes a long way. We are now tackling this project, and would like to report when we succeed.

2. The authors found that the K_m value of substrate myo-inositol is very large ($\sim 12 \pm 2$ mM). The possible reason may be a derivative(s) of myo-inositol which is the real substrate. It would be nice to at least suggest a few possibilities, although experimental validation may be beyond the scope of this work.

We suggested some candidates for the genuine acceptor of TM0415 at the 13th line on page 4. Their chemical structures are shown in **Supplementary Figure 1b**. These compounds are related with the myo-inositol metabolic pathway that is composed of enzymes encoded by the TM0411–TM0416 operon in *Thermotoga maritima*. The compounds have many hydroxyl and carbonyl groups, and thus possibly bind to the acceptor-binding site of TM0415. We did not perform experimental validation because the compounds are not commercially available.

3. The 5-residue “signature” motif is an important result. Have the authors considered to see if it also works on all known PPi kinases beyond ribokinase family?

We investigated whether the signature patterns of TM0415 exist in the known PPi-dependent kinases (PPi-PFK, PPi-PPDK, and PPi-ACK) and added a new paragraph in Discussion on page 10. No signature patterns were observed in PPi-PPDK and PPi-ACK, while PPi-PFKs possess two basic residues in the phosphate-donor-binding site, which are not conserved in ATP-PFK. However, their contributions to PPi binding remain to be elucidated. We think that determination of the PPi-complex structure is indispensable for identifying a signature pattern for the known PPi-dependent kinases belonging to each family.

4. Size exclusion result of TM0415 showing it is a monomer in solution is not shown. It should be included in Supplementary Information.

We added the result of size exclusion chromatography as **Supplementary Figure 3b**. From the result, the molecular weight of TM0415 in solution is estimated to be 24,000. This value is lower than the theoretical molecular weight of a subunit (about 34,000). One of the possible reasons for the lower value of the estimated molecular weight is that there are some interactions between enzymes and chromatography resin. Based on the analysis, we concluded that the most reasonable estimation of the TM0415 oligomeric state is monomer in solution.

5. It is not clear to this reviewer why there is a separate box in Fig. 3D corresponding to AGK_PF and AGK_TL (by the way, “_” is missing).

In order to clarify the meaning of the separate box, we modified the caption of **Figure 3d** and added **Supplementary Figure 7**. As shown in **Supplementary Figure 7**, the GXGD motifs of AGK_PF and AGK_TL are located after a helix highlighted orange (AGKs, lower cartoon), whereas the motif of a typical enzyme of the ribokinase family is followed by the orange helix (upper cartoon). The difference in the order of the magenta and orange elements prevents primary sequence alignment, because primary sequence alignment programs adjust the conserved amino-acid residues by gap insertions, but such insertions cannot change the order of the elements. The separate box in **Figure 3d** highlights the different order of secondary structures of AGK_PF/AGK_TL from those of the other members of the ribokinase family.

We could not find the missing underline, although we thoroughly checked our manuscript. The missing underline pointed by this reviewer might be related with the figure quality. We will provide a higher-quality figure (eps formatted) after the final acceptance of the manuscript.

In addition to the changes based on the reviewers' comments, we made some modifications to our manuscript in order to follow the journal style and clarify the meaning of the sentences. All changes are highlighted in the marked copy of the revised manuscript.